# Human Brain Organoids: A New Model to Study *Cryptococcus neoformans* Neurotropism

**DOI:** 10.3390/jof11070539

**Published:** 2025-07-19

**Authors:** Alfred T. Harding, Lee Gehrke, Jatin M. Vyas, Hannah Brown Harding

**Affiliations:** 1Institute for Medical Engineering and Science, Massachusetts Institute of Technology, Cambridge, MA 02139, USA; alfred.tharding@gmail.com (A.T.H.); lgehrke@mit.edu (L.G.); 2Department of Microbiology, Harvard Medical School, Boston, MA 02115, USA; 3Division of Infectious Diseases, Department of Medicine, Massachusetts General Hospital, Boston, MA 02114, USA; 4Department of Medicine, Harvard Medical School, Boston, MA 02115, USA; 5Broad Institute of MIT and Harvard, Cambridge, MA 02142, USA; 6Division of Infectious Disease, Department of Medicine, Columbia University Vagelos College of Physicians and Surgeons, New York, NY 10032, USA

**Keywords:** cerebral organoid, fungal pathogen, model system, *Cryptococcus neoformans*

## Abstract

With the rise in immunocompromised individuals and patients with immune-related comorbidities such as COVID-19, the rate of fungal infections is growing. This increase, along with the current plateau in antifungal drug development, has made understanding the pathogenesis and dissemination of these organisms more pertinent than ever. The mouse model of fungal infection, while informative on a basic scientific level, has severe limitations in terms of translation to the human disease. Here we present data supporting the implementation of the human cerebral organoid model, which is generated from human embryonic stem cells and accurately recapitulates relevant brain cell types and structures, to study fungal infection and dissemination to the central nervous system (CNS). This approach provides direct insight into the relevant pathogenesis of specific fungal organisms in human tissues where in vivo models are impossible. With this model system we assessed the specific brain tropisms and cellular effects of fungal pathogens known to cross the blood–brain barrier (BBB), such as *Cryptococcus neoformans*. We determined the effects of this fungal pathogen on the overall gross morphology, cellular architecture, and cytokine release from these model organoids. Furthermore, we demonstrated that *C. neoformans* penetrates and invades the organoid tissue and remains present throughout the course of infection. These results demonstrate the utility of this new model to the field and highlight the potential for this system to elucidate fungal pathogenesis to develop new therapeutic strategies to prevent and treat the disseminated stages of fungal diseases such as cryptococcal meningitis.

## 1. Introduction

Invasive fungal pathogens pose an increasing risk to the growing population of immunocompromised individuals and patients worldwide, especially ubiquitous environmental yeasts like *Cryptococcus neoformans* [1,2]. This fungal pathogen has made its environmental reservoir in tree soil and pigeon guano, making it an extreme and ever-present risk to those with compromised immune systems. Furthermore, *C. neoformans* is a professional pathogen that can adapt to the various environments of the host to establish a severe and life-threatening infection, killing 600,000 people annually [1]. One of these environments is that of the human brain. Following inhalation into the lungs, *C. neoformans* traverses the blood–brain barrier (BBB) and persists in the CNS. This brain infection, or meningococcal encephalitis, is extremely difficult to treat due to the lack of new or affordable fungicidal drugs and the inherent resistance of *C. neoformans* to one of the three classes of available antifungals: echinocandins. Annually, it is estimated that one million people develop cryptococcal meningitis, and that over 60% of these individuals succumb to the disease within a few months of diagnosis [3]. Considering the extremely high mortality rate of these infections, it is vital that we further study the interactions between this pathogenic fungus and the human brain in order to enhance patient outcomes following this lethal stage of cryptococcal disease.

Understanding the mechanisms by which this pathogen traverses the BBB has intrigued the field, and many groups have established in vitro models and cell lines for elucidating how *C. neoformans* yeast enters the brain parenchyma from the blood vessel lumen. From these studies, it is thought that *C. neoformans* cells usurp macrophages via a Trojan horse mechanism to cross through the tight junctions and subsequently escape from the macrophage once safely within the brain environment [4]. Others propose that the yeast cells undergo transcytosis themselves to enter the central nervous system without the need for other host cells [5,6]. While this research has been very informative, our understanding of what occurs after traversal that leads to the establishment of infection is severely limited. While mouse models of *C. neoformans* brain infections have yielded insights, some have highlighted the limitations and often lack of translation of animal models to the clinic [7]. To date, the only human models of brain infections remain the use of human cadavers from individuals infected with cryptococcomas, which provide excellent insights into late-stage fatal disease, but do not aid in our understanding of the infection’s initiation or progression [8].

To address these shortcomings, we present here the first use of cerebral organoids derived from human embryonic stem cells (hESCs) as a model for studying the development of fungal disease in the brain; hESCs are a useful platform to generate models of human tissues of interest, including the lungs, liver, kidneys, intestine, and, importantly for this study, the brain. Organoid models derived from human primary cells have been shown to reliably reproduce findings that either match patient data or correlate with clinical trials [9]. The data presented here provide evidence for successful infection of human brain organoids with multiple pathogenic fungi (i.e., *C. neoformans* and *Candida auris*), while also revealing a lack of infection and functional consequence of exposing these organoids to non-pathogenic fungi (i.e., *Saccharomyces cerevisiae*). Furthermore, the tissue-penetrating fungal infection caused by *C. neoformans* not only affects gross morphology and cellular organization/viability, but also generates an immune response, evident through increased cytokine production and an inflammatory transcript response indicative of infection. Taken together, this work demonstrates the potential utility of human organoid models as a strategy for studying fungal-induced pathogenesis.

## 2. Results

### 2.1. Organoids Display Growth Defects in Response to Pathogenic Fungi

To determine whether human cerebral organoids can be infected by the neurotropic pathogenic fungus *C. neoformans*, we co-cultured 21-day old cerebral organoids with *C. neoformans* at two different fungal cell/organoid cell ratios: 0.1:2 (low) and 1:2 (high). On days 7, 14, and 21, we imaged the organoids and measured their size changes by diameter length over time compared to the uninfected control (Figure 1A). By day 7, regardless of the amount of *C. neoformans* added, the infected organoids were significantly smaller compared to the controls. This size reduction was similar to that of organoids infected with neurotropic viruses, often indicating a strong disruption of cellular organization and the induction of cytopathology [10]. Unlike viruses, however, fungal organisms can rapidly grow in cell culture media, consuming the nutrients that cerebral organoids need, and potentially inducing growth defects via nutrient deprivation. To ensure that our observed phenotype was a function of pathogenicity and not simply the presence of a fast-growing fungal organism, we also examined organoid growth over time in the presence of another pathogenic fungus known to infect the brain, *Candida auris* [11], as well as a not traditionally pathogenic fungal organism, *Saccharomyces cerevisiae* (the baker’s yeast). Excitingly, the pathogenic *C. auris* strain also induced a dramatic reduction in organoid growth, similar to *C. neoformans*, while the growth rate of organoids co-cultured with the non-pathogenic *S. cerevisiae* was not significantly different from that of uninfected controls at the end of the infection (day 21), despite an early growth deficit (Figure 1B). These infection experiments demonstrate that human fungal pathogens like *C. neoformans* and *C. auris* can infect human brain organoids and cause significant growth defects by day 7 post-infection, as opposed to non-pathogenic strains like *S. cerevisiae*, which do not induce the same growth defects.

### 2.2. C. neoformans Effectively Penetrates Organoid Tissues

While the growth defects that these organoids present in response to the presence of *C. neoformans* are striking, we next needed to determine whether this fungus was in fact penetrating the organoid tissues to establish infection. Using a periodic acid–Schiff (PAS) stain to detect polysaccharides in tissues, we determined that after 7 days of a *C. neoformans* challenge, organoids had large amounts of polysaccharides present within their tissue layers compared to an uninfected control (Figure 2A (magenta)). To assess this fungal penetration at the species level, we also imaged these organoids using an antibody against the main component of the *C. neoformans* polysaccharide capsule, glucuronoxylomannan (GXM). Using confocal microscopy, we imaged *C. neoformans*-infected organoids at days 7, 14, and 21 and visualized a time-dependent infiltration of GXM-positive fungal organisms into the organoid tissue (Figure 2B (yellow)). This penetration and staining pattern were not seen in the uninfected control, confirming that *C. neoformans* is capable of penetrating into the structure of cerebral organoids without the presence of vasculature. This finding is crucial because it supports our assertion that cerebral organoids could be a useful model for neuroinvasion by fungal pathogens.

### 2.3. C. neoformans Infection Disrupts Organoids’ Cellular Architecture

Having identified both growth defects and invasion of *C. neoformans* into cerebral organoids following infection, we next examined the impacts on organoid cytoarchitecture. During growth, cerebral organoids display a very consistent architectural pattern wherein neural progenitor cells (NPCs) arrange into ventricular patterns and asymmetrically divide outwards to produce neurons and other neural cell types of the brain. Examining the earliest timepoint at which we saw growth defects, 7 days post-infection, we stained organoid sections with antibodies against the transcription factor SOX2 and mitogen-associated protein 2 (MAP2), to image NPCs and neurons, respectively. Excitingly, at this early timepoint (day 7), we were able to see significant differences in the organization of both NPCs and neurons, likely suggesting that our observed growth defects were due to a significant disruption of the standard function and division of both neurons and NPCs. Following a *C. neoformans* infection, both MAP2- and SOX2-positive cells are very sparse and dispersed, and they do not resemble the organization patterns of the uninfected organoid (Figure 3). Furthermore, in an uninfected organoid, the MAP2-positive neuronal cells surround the developing ventricle, but in the infected organoid, these cells are dispersed throughout and do not establish any ventricle-associated pattern. Therefore, this *C. neoformans* infection completely disrupts the typical organization and neuronal structure of these organoids.

### 2.4. Cryptococcal Infection Induces Cytokine Induction in Brain Organoids

A key outcome of cryptococcal CNS infection is the subsequent induction of a wide array of cytokines [12,13,14,15,16,17,18,19,20,21], which are major contributors to disease progression. Importantly, however, not all cells can produce every cytokine. The major cells identified in human cerebral organoids are NPCs, neurons, oligodendrocyte precursor cells, and astrocytes [22]. When comparing the cytokines that these cells are capable of producing with the cytokines known to be induced by cryptococcal infections in the brain, a small list is generated: CXCL10, IL6, CCL5, and IFN-γ. Therefore, we decided to assess the induction of these cytokines in the lysates of our infected and uninfected organoids. Following a 7-day infection with both low and high doses of *C. neoformans*, we were able to detect increased levels of all four cytokines by immunoblot analyses (Figure 4). These same cytokines were not found in the lysates of uninfected organoids, providing further evidence that these fungal pathogens not only infect these tissues but also trigger an inflammatory response in a physiologically relevant manner. These proteins were not detected in the lysates of organoids infected with *C. neoformans* on day 3 (see Geo number GSE302166 for raw data), indicating that the induction of these cytokines is time-dependent. We did not assess protein induction past day 7, as we found the protein of these infected organoids to be completely degraded and unstable for analysis, likely due to the severe disorganization of the organoids themselves, as highlighted in Figure 3.

### 2.5. Organoid Transcriptional Response to Fungal Infection

While understanding the ability of this persisting cryptococcal infection to stimulate the release of specific inflammatory cytokines is important, it does not reveal the full immune response elicited. Therefore, we performed bulk RNA sequencing on the RNA extracted from organoids infected at days 3 and 7 with both our organism of interest, *C. neoformans* (both high and low doses), and another neurotropic fungal pathogen that we found to affect organoid growth over time, *C. auris* (Figure 1B). Importantly, when we attempted this sequencing for day 14 or 21 post-infection, the RNA was determined to be unstable and degraded for organoids that had been infected with both *C. neoformans* and *C. auris*.

To evaluate our results in an unbiased manner, we turned to gene set enrichment analysis (GSEA). Compared to uninfected controls at day 3, a low dose of *C. neoformans* led to increased expression of genes involved in pancreas beta-cell regulation and genes that are targets of MYC (Figure 5A). Neurons and beta cells share many of the transcription factors involved in differentiations, as they both originate from the ectoderm, which might explain the increased expression of pancreas beta-cell-regulatory genes that we observed at day 3 as the cells were undergoing stress from infection [23]. MYC targets are expressed early during brain development and demonstrate increased expression later in the context of traumatic brain injury or cognitive disease [24]. Studies have shown that MYC and MYC targets’ expression is tightly regulated under normal brain development, but overexpression can sensitize cells to apoptosis and cell-cycle activation [25,26]. At a high dose of *C. neoformans* at the same timepoint, we did not observe MYC targets; however, there was increased expression of genes involved in protein secretion, potentially upregulating cytokine release in response to infection. Perhaps most important was the shared induction of the unfolded protein response (UPR) pathway, a hallmark of cellular stress and often a precursor to both apoptosis and inflammation (Figure 5A,B). Interestingly, at day 7 in organoids exposed to both low- and high-dose *C. neoformans*, we observed an increase in transcriptional and cell-cycle-regulatory pathways (E2F targets and G2M checkpoint), as well as a strong upregulation of the type I interferon (IFN) response (Figure 5C,D). We also noted upregulation of the IFN response pathway in response to *C. auris*-infected organoids at day 7, albeit at a slightly weaker level than those seen following *C. neoformans* infection (Appendix A). *C. auris* appeared to induce the UPR and apoptosis pathways more than the type I IFN response, perhaps suggesting that *C. auris* infection in the brain is less immunostimulatory and instead characterized by a stronger induction of cellular stress. Given this upregulation of inflammatory signatures at day 7 across our pathogenic fungal-infected organoids, we also examined the transcriptional state of organoids exposed to *S. cerevisiae* at this same timepoint. Excitingly, we did not observe induction of IFN response pathways in our *S. cerevisiae*-infected organoids, indicating that this inflammatory response is pathogen-dependent. We did, however, observe an increase in the E2F targets and G2M checkpoint pathways in response to *S. cerevisiae*, indicating that although this is not a traditional pathogen, the presence of this yeast in the brain still affects the transcription and cell-cycle progression of these neurons and astrocytes. Interestingly, the *C. auris*-infected organoids at days 3 and 7 had approximately 3-fold more pathways turned on in response to infection than those infected with *C. neoformans* at either dose. These pathways included the unfolded protein response, cholesterol synthesis, apoptosis, and other inflammatory responses. Raw Fastq files, as well as the processed count file, can be found on Geo accession viewer at Geo number GSE302166.

## 3. Discussion

The results presented here are the first to describe a *C. neoformans* infection in human cerebral organoids. We have demonstrated that organoids infected with two different doses of *C. neoformans* have significant growth defects over time compared to uninfected controls. Additionally, we have displayed, through two different methods of fungal detection, a time-dependent infiltration of fungal organisms into the tissue of the organoids. Furthermore, these infected organoids display defects in cytoarchitecture, increased cytokine secretion, and a dose- and time-dependent upregulation of inflammatory pathways. We also note that through the addition of *S. cerevisiae* and *C. auris* as non-pathogenic/negative and positive controls, respectively, we discovered that cerebral organoids respond differently to different fungal organisms in terms of both growth and the transcriptional response. Therefore, these findings present a novel and relevant model system for studying the initial interaction and disease progression when fungal cells reach the human cerebral environment and cause life-threatening disease.

Understanding the interaction between neurotropic fungal pathogens like *C. neoformans* and the brain is essential to improving diagnoses, treatment, and patient outcomes in patients who develop these lethal brain infections. In vitro cell lines and various in vivo animal models for studying cryptococcal meningitis have existed for decades and have uncovered important details about the mechanisms of fungal entry into the CNS; however, they are still lacking in terms of translational potential. Many investigators use human brain endothelial cell lines (i.e., hCMEC/D3) to uncover cryptococcal transmigration across the BBB [6,27,28]. These studies have exposed the complex and multifaceted ways in which *C. neoformans* reaches the CNS through the BBB. There is evidence to suggest both a passive transcytosis across the endothelial cell layer and an active “Trojan horse” phagocyte-dependent crossing of engulfed *C. neoformans* yeast in macrophages through tight junctions between endothelial cells [4,27]. These detailed studies are made possible by the highly controlled environment of working with in vitro cell lines, and the field has progressed due to these findings; however, they only tell part of the pathogenesis story and reveal little about what occurs once the yeast interacts with the complex brain environment. Therefore, other research groups use mouse, rat, rabbit, or zebrafish vertebrate models to investigative the disease progression of *C. neoformans* in the brain. The transparency of zebrafish larvae, the tractable nature of rabbit CNS infections, the natural susceptibility of rats to *Cryptococcus*, and the genetic flexibility of mouse models are just a few of the positives that using these vertebrates provides [7]. However, the lack of clinical translatability and the high cost of maintenance have interfered with research progress, specifically with regard to elucidating the dissemination of *C. neoformans* to the CNS. Additionally, some investigators attempting to study the pathogenesis of this infection using a clinically relevant model for studying cryptococcal brain infections dissect cadavers of human patients who have succumbed to this disease. While this approach has revealed interesting aspects about the pathology of this late and terminal stage of cryptococcal disease in actual human patient brain sections [8], this does not investigate the early interactions between fungal cells and the host brain environment. Therefore, a similarly relevant but less invasive method for studying cryptococcal meningitis is essential. By using human-derived cerebral organoids as a model to study *C. neoformans* infection, we are able not only to study disease initiation and progression amongst clinically relevant cell types but also to quickly process meaningful quantities of samples to reveal key insights into the cellular, cytokine, and transcriptional responses that this infection evokes in the human brain environment at various timepoints throughout its pathogenesis.

The cytokine response uncovered in organoids infected with both low and high doses of *C. neoformans* reveals interesting insights into the behavior of astrocytes, glial cells, neurons, and endothelial cells in response to fungal pathogens. The increased secretion of IL-6 and CXCL10 is particularly exciting because of previous studies connecting these cytokines not only to neuroinflammation but also to fungal disease. IL-6 is repeatedly described as highly induced in the CNS during times of neuroinflammation, such as viral meningitis, murine cerebral malaria, systemic lupus, and HIV-1 [29,30,31,32]. The cell types that secrete IL-6 (neurons, astrocytes, and endothelial cells) are all present in the human cerebral organoids in our model system, making the detection of this cytokine highly likely during neuro-stress and infection [29]. Furthermore, cryptococcal researchers have demonstrated that without IL-6, there is increased permeability of the BBB during fungal meningitis, indicating that this *C. neoformans*-induced IL-6 is integral to preventing fungal migration from the periphery to the CNS [12,13]. Similarly, the pro-inflammatory chemokine CXCL10 has been consistently reported as upregulated in the CNS in response to infection with neurotropic viruses like herpes, HIV-1, hepatitis, and West Nile virus [33]. CXCL10 is also induced in response to cerebral malaria and trypanosome infection [34]. It is believed that this increased chemokine response is essential for mediating the influx of inflammatory leukocytes into the CNS during neuroinflammation through binding its receptor, CXCR3. In the context of cryptococcal meningitis, CXCR3 is essential for lethal brain pathology but dispensable for pathogen clearance [21]. This study examined mice infected with *C. neoformans* and identified the CXCL10-CXCR3 axis as important for the recruitment of T cells to the CNS, similar to what has been demonstrated with viral infection [21]. Furthermore, CXCL10-producing cell types like neurons, astrocytes, and endothelial cells were present in our *C. neoformans*-infected organoids. We further discovered upregulation of IFN-γ in response to *C. neoformans* infection. CXCL10 is also known as IFN-γ-induced protein 10; therefore, this induction could be related to the CXCR3 axis response. Importantly, IFN-γ has been linked to increased fungal cell clearance in a patient population with HIV-associated cryptococcal meningitis, and in a murine model of infection [35,36]. CCL5, which was also induced in our infected organoids, is an important chemokine produced by astrocytes in response to inflammation, especially in diseases such as multiple sclerosis and intracerebral hemorrhage [37,38]. However, much of this induction is essential for microglial recruitment [37], which is something that our organoid model currently lacks. Therefore, it could be that *C. neoformans* infection induces astrocyte secretion of CCL5 to then interact with the receptor CCR5 and recruit various inflammatory cell types like microglia and monocytes. Future studies that incorporate these cell types into our existing organoids will reveal a role for increased CCL5 in the response to *C. neoformans*-induced cerebral damage.

The transcriptional upregulation of specific pathways in response to varied fungi in a dose- and time-dependent manner, as revealed by bulk RNA sequencing, is especially intriguing because it not only exposes the cellular reaction to infection but also reveals a fungal pathogen-specific regulatory response. What is particularly interesting is the upregulation of type I IFN responses in the *C. neoformans*- and *C. auris*-infected organoids at day 7, which was not present in the *S. cerevisiae*-infected group. This fungal pathogen-specific type I IFN response has excited the field lately, as many have connected diverse fungal infections to type I IFN signaling pathways and functional responses [39,40,41,42,43,44,45,46]. A recent finding even identified this IFN response as fungal pathogen-specific and connected the degree of type I IFN signaling to the ability of host cells to endocytose diverse fungal extracellular vesicles (EVs) [47]. The fact that we have identified an upregulation of type I IFN signaling in response to specific fungi in this model, but not in others, begs the following question: might fungal EVs be involved in eliciting this specific response? Future studies investigating not only EVs but other fungal ligands (B-glucan, mannan, capsule, melanin, etc.) as the stimuli that activate these neuronal cell types will elucidate the physical interaction that results in this pro-inflammatory phenotype of fungal-infected cerebral organoids.

Overall, these data establish the use of a novel model system for studying the neuro-pathogenesis of fungal pathogens. This approach provides direct insight into the relevant pathogenesis of specific fungal organisms in human tissues where in vivo models are impossible and in vitro models are lacking. Human organoid models have been used to study a variety of both viral and bacterial pathogens and represent a promising avenue for research of fungal pathogens that disseminate and cause disease in the brain. Our findings support the ability to use this model to assess the specific brain tropisms and cellular effects of fungal pathogens that are known to cross the BBB, such as *C. neoformans*. These insights will allow us to better understand the effects of fungal organisms on the function of the human brain and help us develop treatment strategies for the terminal stages of these fungal diseases, such as cryptococcal meningitis.

## 4. Methods

### 4.1. Organoid Generation from Human Embryonic Stem Cells (hESCs)

Human embryonic stem cells were generated as previously described [48]. Briefly, WIBR3 cells were plated into ultra-low attachment round-bottomed 96-well plates to generate single embryoid bodies (EBs). The EBs were maintained in these plates for approximately 6 days before being exposed to neural induction medium. Between 4 and 6 days after the addition of the neural induction medium, the EBs were embedded in droplets of Matrigel and cultured in neural maturation media in stationary dishes for 4 more days. The organoids were then transferred to an orbital shaker and rotated continuously at 80 rpm for the remainder of the experimentation. Approximately 35 days after formation, the organoids were considered mature and exposed to fungal pathogens or maintained as untreated controls.

### 4.2. Fungal Infection of Organoids

Following the organoids’ growth and maturation for 21 days, fungal organisms were added to the wells containing the growing organoids. Overnight cultures of wildtype *C. neoformans* (H99), *Saccharomyces cerevisiae* (S288C), and *Candida auris* (AR387) were washed 3 times with 1x phosphate-buffered saline (PBS), counted on a LUNA cytometer device, and added at specific ratios to the organoids: 0.1:2 fungal cell/organoid cell ratio for “low” *C. neoformans*, and 1:2 fungal cell/organoid cell ratio for “high” *C. neoformans* and for the controls *S. cerevisiae* and *C. auris*. The organoids were maintained under shaking at 150 rpm in low-adherence 12-well plates at 37 °C and 5% CO_2_, and the medium was changed every 3 days. Uninfected organoids were given fresh maturation media on day 0 and refreshed every 3 days in the same way.

### 4.3. Growth Measurements

Prior to infection, and at pre-determined timepoints (7, 14, 21), images of each organoid were taken on an epifluorescence scope at 4× magnification to ensure that the entirety of the organoid and its diameter was visible. The pictures were processed, and the diameters were measured in FIJI 2 (ImageJ2 version 2.3.0/1.53q). Graphs and statistical analysis (2-way ANOVA with Dunnett’s multiple comparisons test between infected and uninfected at each timepoint) for growth over time were carried out in Prism 10 (* *p* = 0.0314, *** *p* = 0.0005, **** *p* < 0.0001).

### 4.4. Organoid Collection for Staining, Embedding, and Sectioning

On the day of collection, organoids were isolated individually into sterile tubes. The remaining media were aspirated, and the organoids were incubated in 4% paraformaldehyde (PFA) for 30 min at room temperature (RT) to fix the cells. PFA was aspirated and the organoids were washed 5 times with 1x PBS. After the final wash, 15% sucrose in PBS (weight/volume) was added to the organoids, and they were incubated overnight at 4 °C. The next day, 15% sucrose was aspirated, and 30% sucrose in PBS (weight/volume) was added. The tube was then parafilmed to prevent evaporation and stored at 4 °C until all samples were collected. The organoids were then embedded in Optimal Cutting Temperature (OCT) compound and frozen in liquid-nitrogen-cooled 2-methyl-butane. Organoid blocks were stored at −80 °C and cryosectioned using a Leica CM1850 Cryostat (Leica, Wetzlar, Germany). Sections were placed on microscope slides and allowed to dry at RT before storing at −80 °C until staining.

### 4.5. Immunofluorescence Staining

Organoid sections were warmed to RT and then washed in PBS before permeabilization (0.1% Triton X-100 in PBS) and blocking (3% normal goat serum in PBS), both of which occurred simultaneously for one hr. The primary antibodies used included a rabbit monoclonal anti-SOX2 (1:500, Cell Signaling Technology, Danvers, MA, USA), chicken polyclonal anti-MAP2A/B (1:10,000, EnCor Biotechnology, Gainesville, FL, USA), and mouse anti-GXM clone 18B7 (1:1000, Millipore Sigma, Burlington, MA, USA) in blocking buffer at 4 °C overnight. The samples were then washed three times in PBST (0.25% Triton-X). Secondary Alexa Fluor-conjugated goat anti-mouse, anti-chicken, and anti-rabbit antibodies (Invitrogen, Waltham, MA, USA) were diluted 1:1000 and added to blocking buffer with DAPI (1 µg/mL, Invitrogen) and incubated at RT for 1 h. The samples were then washed three times with PBST before being mounted with #1.5 thickness coverslips using ProLong Diamond antifade mounting reagent (Thermo Fisher Scientific, Waltham, MA, USA). Images were captured on a Nikon Eclipse Ti-E Inverted Microscope equipped with a CSU-X1 confocal spinning disk head (Yokogawa, Tokyo, Japan), and a Coherent 4 W continuous-wave laser excited the sample. A 20× high-numerical-aperture objective was used. Images were obtained using an EMCCD camera (Hamamatsu Photonics, Hamamatsu, Japan). Image processing was performed using FIJI 2 (ImageJ version 2.3.0/1.53q).

### 4.6. Fungal Periodic Acid–Schiff Stain

Periodic acid–Schiff staining was performed following the standard kit procedure (Abcam, Cambridge, UK). Cryosection samples were washed in PBS and tap water before being immersed in the periodic acid solution for 10 min. The slides were then washed with distilled water 4 times before being immersed in the Schiff’s solution for 30 min. The slides were then rinsed with hot water and distilled water before staining in light green solution for 2 min. The slides were rinsed with absolute alcohol and then dehydrated with two more rinses before being mounted with #1.5 thickness coverslips. Images taken at 10× or 40× on a Nikon eclipse TS100 epifluorescence microscope (Nikon, Tokyo, Japan) and obtained using an Excelis camera (ACCU-SCOPE Inc., Commack, NY, USA) were processed using FIJI 2 (ImageJ version 2.3.0/1.53q).

### 4.7. Immunoblot Analysis

Lysates were collected in sterile tubes after 7 days of co-cultured incubation at 37 °C and 5% CO_2_. The growth media were aspirated, and accutase was used to break up the tissue structure by incubating for 10 min at 37 °C and 5% CO_2_. The cells were broken up into single-cell suspensions via pipetting and collected via centrifugation at 1000× *g* for 5 min. The lysates were collected using mammalian protein extraction reagent lysis buffer (MPER Thermo Scientific, 78501, Waltham, MA, USA) with sodium orthovanadate and protease inhibitors. The lysates were spun down (14,000× *g* for 5 min at 4 °C), transferred to a fresh tube, and subsequently mixed with 4× NuPage lithium dodecyl sulfate loading buffer and 10× NuPage reducing agent. Western blot assays were performed using a 4–12% NuPage gel, with 2-[N-morpholino] ethanesulfonic acid running buffer (NuPage gels, Thermo Fisher Scientific, Waltham, MA, USA), and transferred to methanol-activated polyvinylidene difluoride membranes (Perkin Elmer, Waltham, MA, USA) using transfer buffer (0.025 M Tris, 0.192 M glycine and 20% methanol) and electrophoretic transfer at 100 V for 1 h. For detection of proteins, polyvinylidene difluoride membranes were blocked for 1 h at RT in 5% milk in TBS 0.01% Tween 20 (TBST). To detect the CXCL10 (Cell Signaling 14969), CCL5 (Cell Signaling 2988), and IL-6 (Cell Signaling 12153) proteins, blots were incubated for overnight at 4 °C in PBST, 1% BSA and primary antibody (1:1000). Following incubation with the primary antibody, the blots were subsequently washed in TBST 3× and incubated with secondary swine anti-rabbit horseradish peroxidase-conjugated antibody at 1:2000 (Agilent DAKO, P0399) (Jackson ImmunoResearch, West Grove, PA, USA) in 1% milk in TBST for 1 h at RT. To detect total protein via actin, blocked blots were incubated for 1 h at RT with simultaneous probing for actin (Cell Signaling 4967L; 1:2500) and secondary swine anti-rabbit horseradish peroxidase-conjugated antibody. The membranes were washed 3× and then visualized using Western Lightning Plus ECL chemiluminescent substrate (Perkin Elmer) on Kodak BioMax XAR film (MilliporeSigma, Burlington, MA, USA). The films were then scanned and processed using Adobe Illustrator version 29.4. Any contrast adjustments were applied evenly to the entire image and adhered to standards set forth by the scientific community.

### 4.8. RNA Isolation and Sequencing

On the day of collection, organoids were isolated individually into sterile tubes. The remaining media were aspirated, and the organoids were incubated in 200 µL of accutase for 10 min at 37 °C and 5% CO_2_ (pipette dup and down to break them up into single-cell suspensions). The cell suspensions were then pelleted at 1000× *g* for 5 min and resuspended in RNA lysis buffer (RLT buffer + β-mercaptoethanol (BME)). RNA was then further extracted from the organoid cells using the Qiagen RNeasy minikit, according to standard protocols. Samples were then submitted to the MIT BioMicrocenter, and total RNA was sequenced using the paired-end Illumina NovaSeq 6000 platform. Raw reads were then run through the RNA-Seq Nextflow pipeline (revision 3.16.0), using the hg38 assembly to generate differential expression data (Deseq2) and gene set enrichment analysis (GSEA) for the described comparisons. Raw Fastq files, as well as the processed count file, can be found on Geo accession viewer at Geo number GSE302166.

## Figures and Tables

**Figure 1 jof-11-00539-f001:**
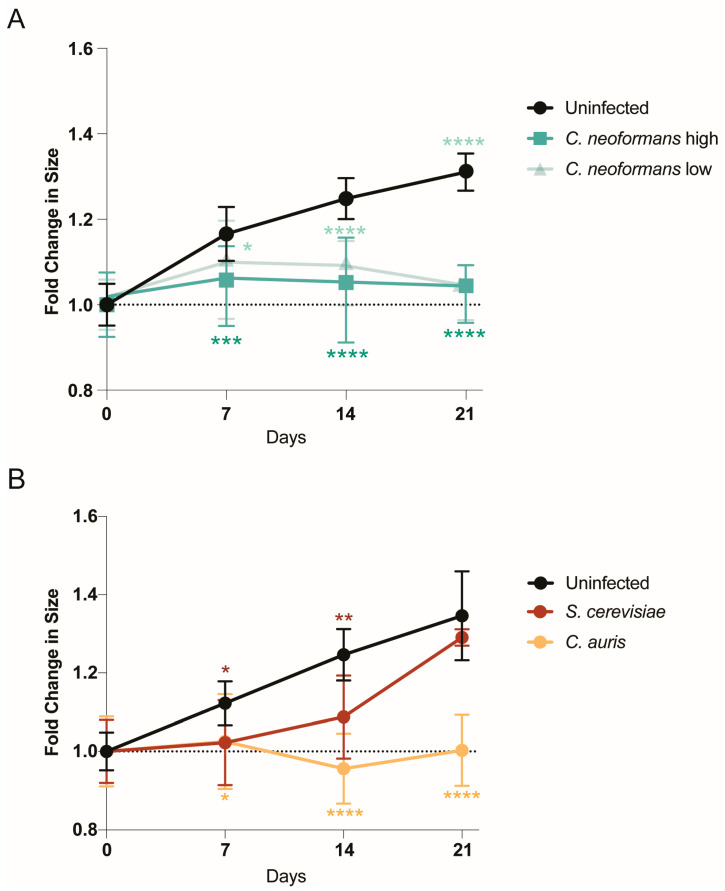
Organoids infected with fungal pathogens decrease in size: (**A**) Diameters of mature organoids infected with high and low doses of *C. neoformans* were measured at the largest cross-section on days 0, 7, 14, and 21. These diameters were compared to those of an uninfected control at the same timepoints, and the fold change in size was plotted with the mean and SD. A minimum of n = 4 organoids were measured for each timepoint. Significance was assessed by an ordinary one-way ANOVA and Dunnett’s multiple comparisons test: * *p* = 0.0314, *** *p* = 0.0005, and **** *p* < 0.0001. (**B**) Diameters of mature organoids infected with *S. cerevisiae* and *C. auris* were measured at the largest cross-section on days 0, 7, 14, and 21. These diameters were compared to those of an uninfected control at the same timepoints, and the fold change in size was plotted with the mean and SD. A minimum of n = 4 organoids were measured for each timepoint. Significance was assessed by an ordinary one-way ANOVA and Dunnett’s multiple comparisons test: * *p* = 0.0128, ** *p* = 0.0079, and **** *p* < 0.0001.

**Figure 2 jof-11-00539-f002:**
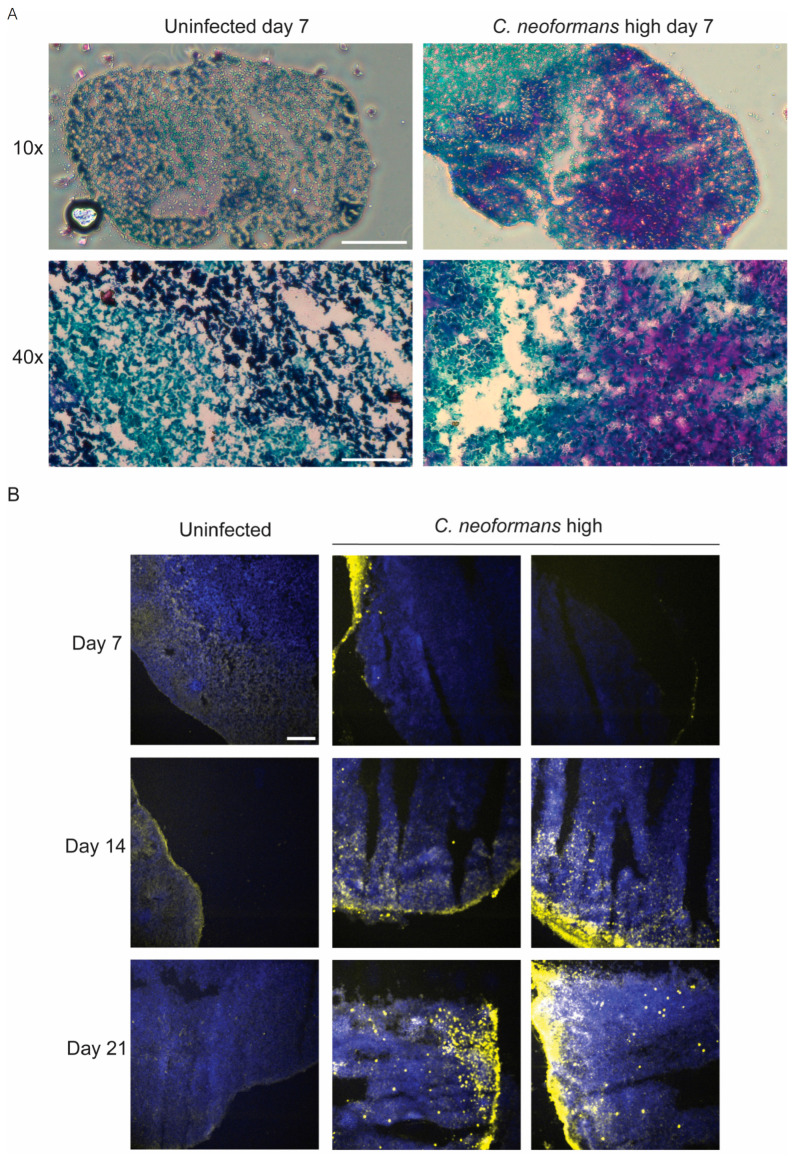
Organoids have *C. neoformans* present throughout infection: (**A**) Periodic acid–Schiff (PAS) stain to detect polysaccharides in organoid sections infected with a high dose of *C. neoformans* compared to an uninfected control at day 7. The blue-stained sections at both 10× and 40× represent tissue; magenta-stained sections represent fungal cell walls (polysaccharides and glycogen). Scale bar for 10× = 100 µm; scale bar for 40× = 25 µm. (**B**) Confocal microscopy images of sections of organoids infected with a high dose of *C. neoformans* compared to an uninfected control at days 7, 14, and 21. Sections were incubated with the anti-GXM clone 18B7, and subsequently with the secondary Alexa Fluor-conjugated goat anti-mouse with DAPI. The blue represents nuclei that are positive for DAPI, and the yellow indicates areas that are positive for GXM (fungal capsule polysaccharide). Scale bar = 50 µm.

**Figure 3 jof-11-00539-f003:**
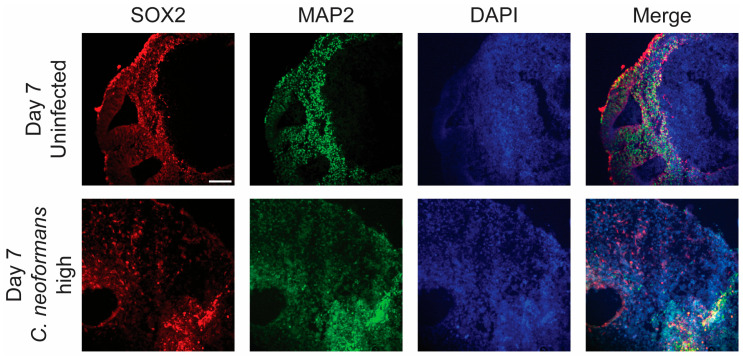
Organoids decreased neural progenitor cells and neurons after infection: Confocal microscopy panels of organoid sections infected with a high dose of *C. neoformans* at day 7 compared to an uninfected control. Sections were stained with antibodies against the NPC transcription factor SOX2 (red), the neuronal marker MAP2 (green), and the nuclear marker DAPI. The merged images reveal organoid architecture and organization in uninfected mature organoids and a disordered state in the infected organoids. Scale bar = 50 µm.

**Figure 4 jof-11-00539-f004:**
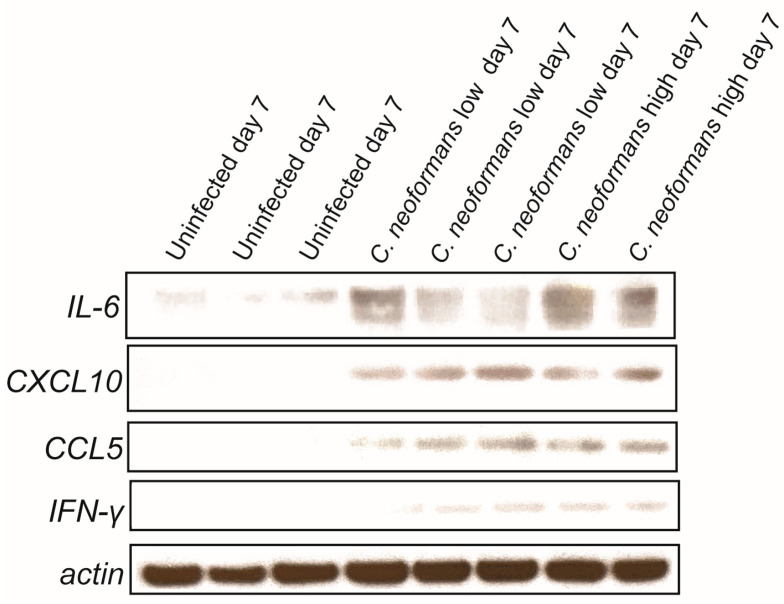
*C. neoformans*-infected organoids have increased cytokine expression and upregulated pro-inflammatory pathways: Immunoblot of IL-6, CXCL10, CCL5, IFN-γ, and actin (loading control) from the lysate of uninfected and *C. neoformans*-infected organoids at both low and high doses at day 7.

**Figure 5 jof-11-00539-f005:**
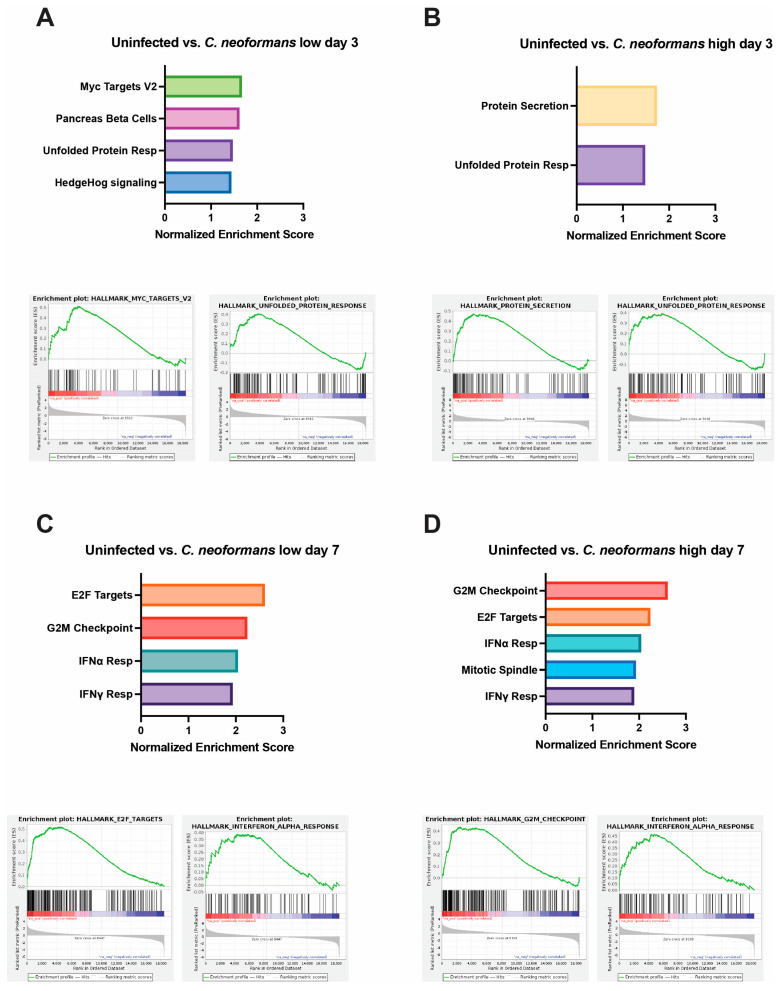
Gene set enrichment analysis of cerebral organoids following infection with *C. neoformans*: (**A**,**B**) Graphed normalized enrichment scores of the hallmark pathways that were significantly enriched (FDR ≤ 0.05), and a set of enrichment plots for select pathways for organoids infected with a low (**A**) and high (**B**) dose of *C. neoformans* 3 days post-infection. (**C**,**D**) Graphed normalized enrichment scores of the hallmark pathways that were significantly enriched (FDR ≤ 0.05), and a set of enrichment plots for select pathways for organoids infected with a low (**A**) and high (**B**) dose of *C. neoformans* 7 days post-infection.

## Data Availability

The original contributions presented in this study are included in the article/Appendix A. Further inquiries can be directed to the corresponding authors. Raw Fastq files, as well as the processed count file, can be found on Geo accession viewer at Geo number GSE302166.

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
