# Peer review of "Human Brain Organoids: A New Model to Study Cryptococcus neoformans Neurotropism"

_jof, 2025, doi:10.3390/jof11070539_

Round 1

Reviewer 1 Report

I thoroughly enjoyed reading the manuscript by Harding et al., which elegantly demonstrates the utility of human brain organoids as a model for studying the neurotropism of Cryptococcus neoformans. The manuscript is, overall, well structured and clearly written. The introduction provides sufficient background to contextualize the study, and the results are presented in a way that allows the reader to follow the progression of the findings. The discussion is concise and effectively highlights the advantages of this novel model for investigating the early stages of infection by Cryptococcus and potentially other neurotropic pathogens.

One area where the manuscript could be strengthened is the inclusion of data on fungal burden at the time points used to assess organoid size. This would help clarify whether the observed reduction in organoid size correlates with increasing fungal load. Additionally, assessing fungal growth in the culture medium, independent of the organoids, could help rule out the possibility that organoid shrinkage is due to nutrient depletion from fungal overgrowth rather than a direct effect of infection.

As a minor suggestion, I recommend unifying the y-axis scales in the graphs of Figure 1 to facilitate easier visual comparison between panels.

In Figure 5, the enrichment plots are not legible and do not convey additional information beyond what is already summarized in the normalized enrichment scores. I suggest moving these plots to the supplementary material and removing them from the main figure. Similarly, the enrichment plots in Figure S1 are also difficult to read; presenting them as separate high-resolution files would improve their usability.

Line 233: The authors state that S. cerevisiae did not induce an IFN response; however, the data supporting this statement are not shown, as Table 1, referenced in the text, is missing from the manuscript. Please ensure this table is included.

Methods Section

The description of the methods lacks consistency in writing style, making the section feel disjointed. For example, section 4.4 is written using both past and present tense. I recommend revising this section for stylistic uniformity and clarity.

Lines 391 and 419: Please standardize the spelling of the software name to ensure consistency throughout the manuscript.

Line 396: The acronyms PFA and RT have not been previously defined. Additionally, "RT" is later spelled out as "room temperature” or “room temp” (e.g., on lines 405 and 413). For clarity, define these acronyms upon first use and maintain consistency in their usage throughout.

Line 400: The acronym OCT is used without prior definition. Please provide the full term upon first mention.

Lines 405–407: It is unclear whether the 1-hour incubation time refers only to blocking or also includes permeabilization. Please clarify this step in the methodology.

Lines 407, 413, 444: Standardize the notation used for indicating time (e.g., “h” for hours, “min” for minutes) throughout the methods section.

Lines 422–423 and 449–450: The phrasing in these lines is unclear. Please revise for clarity and grammatical precision.

Line 452: The catalog number for the anti-actin antibody is not provided. Please include this information for reproducibility.

Lines 461–462: The acronyms RLT and BME are used without prior definition. Please define them when first mentioned.

Author Response

Reviewer 1 major comments:

  • Comment 1: One area where the manuscript could be strengthened is the inclusion of data on fungal burden at the time points used to assess organoid size. This would help clarify whether the observed reduction in organoid size correlates with increasing fungal load. Additionally, assessing fungal growth in the culture medium, independent of the organoids, could help rule out the possibility that organoid shrinkage is due to nutrient depletion from fungal overgrowth rather than a direct effect of infection.
  • Response: We thank the reviewer for this comment. We attempted to answer the viability question by plating CFUs from the supernatant of the wells of the infected organoids at day 0, day 7 and day 14 and observed live yeast at all time points in all infections. We did note that these live yeast in the supernatant decreased over time, which might be confounded by replacing the maturation media every other day in order to maintain organoid viability (which is an essential step). However, decreased fungal growth over time might support the “direct effect of infection” hypothesis the reviewer presents. Another confounding part of this experimental approach is that by only assessing the live yeast in the surrounding media (supernatant), we were not truly measuring live yeast in the infected organoids as we left these untouched or processed them in different ways at these time points (RNA or protein). Therefore, we performed experiments showing fungal growth and invasion over time with neoformans, our main pathogen of interest, and based on the strong results from these experiments, we do not feel the current manuscript needs additional experimentation at this time. Future studies will assess CFU in the organoids themselves.

Reviewer 1 minor revisions:

  • Comment 1: As a minor suggestion, I recommend unifying the y-axis scales in the graphs of Figure 1 to facilitate easier visual comparison between panels.
  • Response: We agree with the reviewer and have edited the Y-axis of Figure 1A to mirror that of Figure 1B to aid in ease of interpretation and comparison.

  • Comment 2: In Figure 5, the enrichment plots are not legible and do not convey additional information beyond what is already summarized in the normalized enrichment scores. I suggest moving these plots to the supplementary material and removing them from the main figure. Similarly, the enrichment plots in Figure S1 are also difficult to read; presenting them as separate high-resolution files would improve their usability.
  • Response: We agree that the resolution of these figure panels is suboptimal and have therefore improved the resolution and legibility in this next version. We thank the reviewer for drawing our attention to this lack of usability. Now that they are higher resolution, we feel they merit inclusion in the main figure of the manuscript as they further support the RNA-seq data.

  • Comment 3: Line 233: The authors state that  cerevisiaedid not induce an IFN response; however, the data supporting this statement are not shown, as Table 1, referenced in the text, is missing from the manuscript. Please ensure this table is included.
  • Response: We thank the reviewers for discovering our error in citing a “table 1” that was not included. We decided to instead upload all Raw Fastq files as well as the processed count file to the Geo accession viewer at Geo number GSE302166. We have removed the appropriate text to ensure no conclusion and updated the methods to indicate where to access this data publicly.
    • Revised MS, Page 8, line 239-241.
    • Revised MS, Page 15, line 473-474.

Methods Section Comments:

  • The description of the methods lacks consistency in writing style, making the section feel disjointed. For example, section 4.4 is written using both past and present tense. I recommend revising this section for stylistic uniformity and clarity.
  • Response: We thank the reviewer for noticing these inconsistencies and have carefully revised the methods section to ensure correct tense and clarity.

  • Lines 391 and 419: Please standardize the spelling of the software name to ensure consistency throughout the manuscript.
  • Response: We have edited the text to standardize the spelling of software (FIJI 2 (ImageJ2 version 2.3.0/1.53q)) throughout.
    • Revised MS Page 13, line 393.

  • Line 396: The acronyms PFA and RT have not been previously defined. Additionally, "RT" is later spelled out as "room temperature” or “room temp” (e.g., on lines 405 and 413). For clarity, define these acronyms upon first use and maintain consistency in their usage throughout.
  • Response: We thank the reviewer for making us aware of these issues. We have gone through the manuscript and edited any first mentioned abbreviations to the full term.
    • Revised MS Page 13, Line 398-399
    • Revised MS Page 13, Line 407
    • Revised MS Page 13, Line 410
    • Revised MS Page 13, Line 418

  • Line 400: The acronym OCT is used without prior definition. Please provide the full term upon first mention.
  • Response: We thank the reviewer for making us aware of these issues. We have gone through the manuscript and edited any first mentioned abbreviations to the full term
    • Revised MS Page 13, line 404

  • Lines 405–407: It is unclear whether the 1-hour incubation time refers only to blocking or also includes permeabilization. Please clarify this step in the methodology.
  • Response: We agree that this wording is confusing. We have reworded this sentence to provide clarity as to the sequence of events for this experiment.
    • Revised MS Page 13, line 411-412

  • Lines 407, 413, 444: Standardize the notation used for indicating time (e.g., “h” for hours, “min” for minutes) throughout the methods section.
  • Response: We thank the reviewer for making us aware of these issues. We have gone through the manuscript and standardized these terms.
    • Hour = hr and minutes = mins throughout.

  • Lines 422–423 and 449–450: The phrasing in these lines is unclear. Please revise for clarity and grammatical precision.
  • Response: We thank the reviewer for pointing out these issues and lack of clear sentence structure, we have edited the lines in the revised manuscript.

  • Line 452: The catalog number for the anti-actin antibody is not provided. Please include this information for reproducibility.
  • Response: We have added this catalog number 4967L and thank the reviewer for noting its absence.
    • Revised MS Page 14, line 456

  • Lines 461–462: The acronyms RLT and BME are used without prior definition. Please define them when first mentioned.
  • Response: We thank the reviewer for making us aware of these issues. We have gone through the manuscript and edited any first mentioned abbreviations to the full term.
    • Revised MS Page 14, line 466-467

Reviewer 2 Report

The manuscript is interesting and a remarkable contribution to the field. As Cryptococcus is an emerging pathogen that becomes more and more frequent nowadays due to an increase in the number of people with immunosuppression, I think novelties related to this yeast are always welcome. As the authors underlined very well, I think this model using organoids could be of great help in understanding better the pathogenesis of this microorganism. 

I think the manuscript is well-written, it fills a gap in literature and it can be published with only minor revisions. I have made some more detailed suggestions below. 

  • please describe the abbreviations the first time they are mentioned in the text (e.g. line 396 - RT, PFA; line 400 - OCT). Please check through the text and correct
  • please check the English language and correct where necessary. There are sentences that don't sound scientific or those that have missing words (e.g. line 431)
  • in the results section, I think the first paragraph belongs more to the material and methods section. Unfortunately, I think eliminating it (or moving it) would make the results very hard to understand since the methods are described at the end. In my opinion articles have a better flow when the materials and methods section is added before the results. Since this is a matter of personal preference, you can take my suggestion into consideration. Otherwise, the structure can remain the same. The content of both sections is well-written and reproducible, so I don't think there is anything to correct regarding content.

Author Response

Dear JoF editorial board,

We thank the editor and the reviewers for their thoughtful and timely comments on our manuscript (jof-3746254). We have addressed all reviewer concerns below and in the revised manuscript in red text. Additionally, we have uploaded our RNA-seq data set to the GEO database and added the accession number (GSE302166) to the methods section of the revised manuscript (page 15, line 469).

Reviewer 1 minor revisions

  • Comment 1: please describe the abbreviations the first time they are mentioned in the text (e.g. line 396 - RT, PFA; line 400 - OCT). Please check through the text and correct
  • Response: We thank the reviewer for making us aware of these issues. We have gone through the manuscript and edited any first mentioned abbreviations to the full term.
    • Revised MS Page 13, Line 396
    • Revised MS Page 13, line 400
    • Revised MS Page 13, line 381

  • Comment 2: please check the English language and correct where necessary. There are sentences that don't sound scientific or those that have missing words (e.g. line 431)
  • Response: We thank the reviewers for catching this mistake on line 431. We have edited the text to correct this and proofread the rest of the text to ensure clear language.
    • Revised MS Page 14, Line 434

  • Comment 3: in the results section, I think the first paragraph belongs more to the material and methods section. Unfortunately, I think eliminating it (or moving it) would make the results very hard to understand since the methods are described at the end. In my opinion articles have a better flow when the materials and methods section is added before the results. Since this is a matter of personal preference, you can take my suggestion into consideration. Otherwise, the structure can remain the same. The content of both sections is well-written and reproducible, so I don't think there is anything to correct regarding content.
  • Response: We agree with this reviewer point that rearranging the structure of the paper would be the best way to solve this issue. As this is out of our control, we have decided to leave the first paragraph of the results as it is written. As the reviewer said, removing this would most likely confuse the reader and these details. while ‘method-heavy’, are necessary to understand the entire results section.

Reviewer 3 Report

Dear authors:

Thank you very much for this article.

Line 74: It does not seem very clear to me to classify Candida auris as a yeast with neurotropism. Although there are cases of C. auris causing meningoencephalitis, they are not the most frequent.

Figure 1 b: Was the fungus's viability measured at days 7 and 14? If it was calculated, how was the viability of the strain measured?
In Figure 1, B, how do you explain that at day 7, S. cerevisiae and C. auris have the same effect on organoid diameter?
How can you rule out that, after day 7, the effect is somewhat due to the death of the S. cerevisiae strain?

Kind regards

line 74: it does not seem very clear to me to classify candida auris as a yeast with neurotropism. Although there are cases of C. auris causing meningoencephalitis, they are not the most frequent.

figure 1 b: was the viability of the fungus measured at days 7, 14? if it was measured, how was the viability of the strain measured?
in figure 1, B, how do you explain that at day 7 S. cerevisiae and C. auris have the same effect on organoid diameter?
how can you rule out that after day 7 the effect is rather due to the death of the S. cerevisiae strain?

Author Response

Dear JoF editorial board,

We thank the editor and the reviewers for their thoughtful and timely comments on our manuscript (jof-3746254). We have addressed all reviewer concerns below and in the revised manuscript in red text. Additionally, we have uploaded our RNA-seq data set to the GEO database and added the accession number (GSE302166) to the methods section of the revised manuscript (page 15, line 469).

Reviewer 2 minor revisions

  • Comment 1: Line 74: It does not seem very clear to me to classify Candida auris as a yeast with neurotropism. Although there are cases of C. auris causing meningoencephalitis, they are not the most frequent.
  • Response: We thank the reviewer for this comment and we agree that this classification is misleading. We have changed the wording in the text as follows:
    • Revised MS Page 2, Line 73: The data presented here provide evidence for successful infection of human brain organoids with multiple pathogenic fungi (i.e., C. neoformans and Candida auris), while also revealing a lack of infection and functional consequence of exposing these organoids to non-pathogenic fungi (i.e., Saccharomyces cerevisiae).
  • Comment 2:
  1. A) Figure 1 b: Was the fungus's viability measured at days 7 and 14? If it was calculated, how was the viability of the strain measured?
    B) In Figure 1, B, how do you explain that at day 7, S. cerevisiae and C. auris have the same effect on organoid diameter?
    C) How can you rule out that, after day 7, the effect is somewhat due to the death of the S. cerevisiae strain?
  • Response:
    • A) We thank the reviewer for these questions. We attempted to answer the viability question by plating CFUs from the supernatant of the wells of the infected organoids at day 0, day 7 and day 14 and observed live yeast at all time points in all infections. However, we did note that these live yeast in the supernatant decreased over time, which was most likely confounded by replacing the maturation media every other day in order to maintain organoid viability (which is an essential step). Furthermore, by only assessing the live yeast in the surrounding media (supernatant), we were not truly measuring live yeast in the infected organoids as we left these untouched or processed them in different ways at these time points (RNA or protein). Future experiments will assess live cerevisiae and C. auris yeast within the organoid itself, similar to what we did with the PAS stain and GXM antibody for the two doses of C. neoformans. However, because we performed these experiments showing fungal growth and invasion over time with C. neoformans, which was our main pathogen of interest, we do not feel the current manuscript needs additional experimentation.
    • B) Regarding the day 7 growth phenotype in the cerevisiae and C. auris infected wells, we do not consider this to be an unexpected result. C. neoformans is a truly neurotropic fungal pathogen and thusly expect it to induce rapid pathogenesis. C. auris, which is capable of infecting the brain, is not considered a neurotropic pathogen, as the reviewer emphasized in comment one. We anticipated that it would likely cause more pathogenesis than S. cerevisiae, which is historically nonpathogenic, but likely cause less or a delayed onset of pathogenesis when compared to C. neoformans. In our opinion, this  “delayed onset” or “moderate” pathogenesis supports the idea that C. auris is not a neurotropic fungus like C. neoformans. We decided to add S. cerevisiae and C. auris as controls in these experiments and feel they provided interesting data as controls, but the true finding is in the C. neoformans infected wells.
    • C) Finally, we thank the reviewer for this question about the viability of cerevisiae and the lack of growth defect of the organoids infected with this yeast at later timepoints. As we included S. cerevisiae as an additional negative control, we did not follow up on the infectivity or invasion of this yeast into the organoid. Future studies interested in using organoids as models to study S. cerevisiae infection will need to investigate the viability of the yeast at later timepoints. We did however see viable yeast in the supernatant of the infected wells late into infection, similarly to C. auris and C. neoformans (as described in response (A)), indicating that the yeast is at least replicating in these culture conditions and could affect organoid growth in this way.